# Temperature Effect on Deformation Mechanisms and Mechanical Properties of Welded High-Mn Steels for Cryogenic Applications

**DOI:** 10.3390/ma17164159

**Published:** 2024-08-22

**Authors:** Minha Park, Gang Ho Lee, Geon-Woo Park, Gwangjoo Jang, Hyoung-Chan Kim, Sanghoon Noh, Jong Bae Jeon, Byoungkoo Kim, Byung Jun Kim

**Affiliations:** 1Energy System Group, Korea Institute of Industrial Technology, Busan 46938, Republic of Korea; minha6931@gmail.com (M.P.); rkdgh546@kitech.re.kr (G.H.L.); dudi9112@kitech.re.kr (G.-W.P.); zoo12052@kitech.re.kr (G.J.); chancpu@kitech.re.kr (H.-C.K.); 2Department of Materials Science and Engineering, Pukyong National University, Busan 48513, Republic of Korea; nohssang@pknu.ac.kr; 3Department of Materials Science and Engineering, Dong-A University, Busan 49315, Republic of Korea; jbjeon@dau.ac.kr

**Keywords:** high-Mn steels, cryogenic temperatures, submerged arc welding (SAW), deformation behaviour, martensite transformation

## Abstract

High-manganese steel (high-Mn) is valuable for its excellent mechanical properties in cryogenic environments, making it essential to understand its deformation behavior at extremely low temperatures. The deformation behavior of high-Mn steels at extremely low temperatures depends on the stacking fault energy (SFE) that can lead to the formation of deformation twins or transform to ε-martensite or α′-martensite as the temperature decreases. In this study, submerged arc welding (SAW) was applied to fabricate thick pipes for cryogenic industry applications, but it may cause problems such as an uneven distribution of manganese (Mn) and a large weldment. To address these issues, post-weld heat treatment (PWHT) is performed to achieve a homogeneous microstructure, enhance mechanical properties, and reduce residual stress. It was found that the difference in Mn content between the dendrite and interdendritic regions was reduced after PWHT, and the SFE was calculated. At cryogenic temperatures, the SFE decreased below 20 mJ/m^2^, indicating the martensitic transformation region. Furthermore, an examination of the deformation behavior of welded high-Mn steels was conducted. This study revealed that the tensile deformed, as-welded specimens exhibited ε and α′-martensite transformations at cryogenic temperatures. However, the heat-treated specimens did not undergo α′-martensite transformations. Moreover, regardless of whether the specimens were subjected to Charpy impact deformation before or after heat treatment, ε and α′-martensite transformations did not occur.

## 1. Introduction

High-manganese (high-Mn) steel, recognized for its exceptional strength and toughness, is a promising material for diverse cryogenic applications, including liquefied natural gas (LNG) tanks, transportation systems, and LNG terminal systems. LNG ships require low temperature systems because LNG is transported at cryogenic temperatures below −163 °C [1,2]. The LNG cryogenic systems require a variety of components, including plates, pipes, and valves, and these components must be connected by a welding process. Especially, welded pipes of various sizes are required, from small diameter to large diameter. To withstand extreme conditions such as low temperatures and high pressures, thick and welded pipes with an outer diameter of 20 to 30 inches or more are required. However, pipes and components manufactured from high-Mn materials are still under development, and much research is still needed to ensure their reliability and performance in cryogenic environments. Therefore, high-Mn steel could be an alternative option for key components in low temperature systems, especially other cryogenic materials. The excellent properties of high-Mn steel also make it suitable for use in parts of the cryogenic system that operate at cryogenic temperatures. 

The high manganese content in the steel allows for improved deformation behavior, which is important for parts that are subject to high stresses and strains. High-Mn steels designated as twinning-induced plasticity (TWIP) steels are known for their austenite single phase at room temperature and various deformation behaviors such as dislocation glide, TWIP, and transformation-induced plasticity (TRIP), depending on the stacking fault energy (SFE) range [3,4,5]. In room-temperature conditions, the welded high-Mn steel forms deformation twins under deformation. These deformation twins effectively subdivided the grain, reducing the average mean free path of dislocations, which is known as the dynamic Hall-Petch effect, which simultaneously increases both strength and toughness as the grain size decreases [6,7]. However, the deformation behavior of high-Mn steels depends on the SFE, which decreases as the temperature decreases [4,8,9]. Below the SFE of 20 mJ/m^2^ [4], TRIP mechanisms are activated instead of TWIP, resulting in martensite transformation where deformation twins intersect [10,11,12]. The complex microstructure resulting from this can potentially have a detrimental impact on the mechanical properties of high-Mn steel. Therefore, understanding the deformation mechanisms is crucial for improving performance in extremely low-temperature applications.

Welding is crucial for the production of structural materials like pipes, vessels, and storage tanks that are used in various industries. Different welding methods, such as laser welding [13], electron beam welding [14] and spot welding [15] are commonly used for welding thin sheets of high-Mn steels due to the low distortion and heat input from the welding process. Recently, submerged arc welding (SAW) has gained attention as an effective method for welding thick pipes or plates in the cryogenic industry [16,17,18]. However, weldment of high-Mn steels has disadvantages, including Mn-segregation, a wide heat affected zone (HAZ) owing to high heat input, and multiple weld passes [19,20]. Especially, Mn-segregation occurs due to variations in the Mn content between dendrites and the interdendritic region in the weldment, potentially impacting the mechanical properties of the high-Mn steel. This segregation is attributed to the evaporation of Mn, which occurs during welding processes at temperatures exceeding 1500 °C [21,22]. And the solubility difference of manganese (Mn) between its solid and liquid phases can further exacerbate this issue during the solidification of the weldment [23,24,25]. The formation of Mn-segregation is typically attributed to the greater Mn solubility in the liquid phase when compared to its solubility in the solid phase [23,26]. Variation in Mn content between the interdendritic region and dendrite can cause differences in SFE, leading to distinct modes of local deformation [27,28,29]. In order to address this issue, post welding heat treatment (PWHT) is used to obtain a homogeneous microstructure, improve mechanical properties, and reduce residual stress [30]. The deformation behavior of welded high-Mn is influenced by SFE, temperature, and loading rate. Therefore, it is crucial to understand it [3,4,5,31,32]. The aim of this study is to examine the effects of PWHT on the mechanical properties of welded high-Mn steels at room and cryogenic temperatures. Additionally, the research aims to explore the effect of deformation temperature on the deformation behavior of welded high-Mn steels.

## 2. Materials and Methods

In this study, the compositions of filler metal were 0.21 wt.% C, 21.5 wt.% Mn, 0.51 wt.% Si, 0.004 wt.% S and 3.33 wt.% Cr. A submerged wire (POS-CF1) with the baked type flux for cryogenic high-Mn steel manufactured by Poswelding (Pohang, Republic of Korea) was used in this study. A 20 mm thick high-Mn steel was manufactured using a hot rolling process following a heat procedure at 1250 °C. In order to manufacture the 20-inch high-Mn steel pipes, they were roll-bent into a diameter of 20 inches, and then SAW welding was applied to the roll-bent pipes. The welding voltage was 28 V, and the welding current ranged from 640 to 747 A. The welding was carried out using a double v-groove through triple-pass, and the welding speed was maintained from 0.45 to 0.52 mm/min. The heat input during the process ranged from 21.5 to 24.3 kJ/cm. And then, the PWHT was implemented at temperatures of 1000 °C for 2 h to enhance the mechanical properties of the welded high-Mn steels, followed by water quenching. These welded specimens are referred to as ‘as-welded’ before heat treatment and ‘W1000’ following heat treatment at 1000 °C. Figure 1 illustrates the schematic diagram of the PWHT process (Figure 1a), and it also depicts the location of the Charpy impact test specimen and the tensile test specimen within the welded high-Mn steel (Figure 1b). 

The x-ray diffraction (XRD) analysis was performed using an X’pert Pro machine (Marven Panalytical’s, Marvern, United Kingdom) equipped with Cu Kα diffraction sources. The XRD pattern was generated by scanning the 2θ range from 40° to 100° at a scan speed of 0.026° per second. To observe phase transformation, the peaks of (111), (200), (220), (311), and (222) austenite phases, as well as (101), (102), and (201) ε-martensite phase, and (110), (200), and (211) α′-martensite phases, were utilized. 

Tensile testing was conducted at 20 °C (room temperature) and –160 °C (cryogenic temperatures) utilizing a 100 kN tensile testing machine (MTS E45) with a strain rate of 10⁻^3^ s⁻^1^. The tensile specimens were fabricated from the center of the welded high-Mn steel, as seen in Figure 1b. The specimens were sub-sized plate-type specimens with a gauge length of 12.5 mm, a width of 3 mm, and a thickness of 1 mm. The ultimate tensile strength (UTS) was identified by the maximum load, and the yield strength (YS) was determined as the load corresponding to 0.2% offset. Additionally, the elongation was calculated by measuring the crosshead displacement. In order to conduct the tensile test at cryogenic temperatures, a specialized chamber was linked to a liquified nitrogen tank, which was then connected to the tensile testing machine. The specimen’s temperature was meticulously regulated using evaporative cooling with liquid nitrogen. 

A charpy impact test was performed at both 20 °C and −160 °C using an automated impact testing machine equipped with a PSW750 pendulum (750 J capacity, Zwick Roell). Charpy impact test specimens were taken from the centerline of the weldment of high-Mn steel, as depicted in Figure 1b. The impact test specimens were fabricated according to ASTM E23 (10 mm in width and thickness, 55 mm in length). In addition, a V-notch was formed within the weldment with a depth of 2 mm. The load–displacement data can be acquired through an instrumented Charpy impact test using an instrumented pendulum (velocity of 5.424 m/s).

To observe the microstructure of welded high-Mn steel, microstructural analysis was performed using optical microscopy (OM, Olympus Corporation, GX51F, Okayama, Japan). Samples for microstructural investigation were prepared by mechanically grinding them using abrasive paper and then polishing them with diamond paste. And then, chemical etching was performed using a 5% nital solution for 30 s. For the microstructure of a deformed tensile specimen, electron backscatter diffraction (EBSD, Oxford instrument, Oxford, UK) analysis was investigated with OIM software (EDAX, TSL OIM Analysis 7, Redwood Shore, CA, USA) by a field emission scanning electron microscope (FE-SEM, Hitachi, Tokyo, Japan) machine. To prepare the EBSD specimens, the surfaces were first removed by grinding them with abrasive paper. Subsequently, the electro-polishing was conducted with an etchant solution (90% acetic acid + 10% perchloric acid). The electro-polishing process was carried out at 20 V for 60 s using an electropolishing machine (Lectorpol-5, Struers, Copenhagen, Denmark). Especially in the case of fractured Charpy impact specimens, electroless Ni plating was performed to investigate the deformation mechanism on the fracture surfaces of the impact specimen. The electroless Ni plating was conducted for 3 h at a temperature of approximately 85 °C using a Ni plating solution. For phase transformation analysis, the fractured Charpy impact specimens were subjected to mechanical grinding using abrasive paper and polishing with diamond paste (using 3, 1, 1/4 μm, and 0.04 μm).

To investigate the tensile deformed microstructure at a temperature of −163 °C, the specimens were cut by focused ion beam (FIB) milling (sample size 10 × 10 μm). Subsequently, the deformed specimens were examined by a field emission scanning transmission electron microscope (FE-STEM, Talos F200X, Thermo Fisher Scientific, Waltham, MA, USA) operating at 200 kV. 

## 3. Results

### 3.1. Microstructural Examination of Welded High-Mn Steel in the As-Welded and PWHT Conditions

Figure 2 shows the microstructure of welded high-Mn steels before heat treatment and heat treatment at 1000 °C. Before heat treatment, the as-welded microstructure exhibited a columnar dendritic structure, as shown in Figure 2a. The dendrite structure tends to grow in the direction of heat flow, which in this case is vertical in the center of the weldment. Additionally, there are dendrites and interdendritic regions in the weldment where segregation occurs during the solidification process. Figure 2b exhibited the microstructure of heat-treated welded high-Mn steel at 1000 °C. It is similar to the as-welded specimen with columnar dendrite structure, but it can be seen that the microsegregation has been reduced by PWHT. During the heat treatment process, the high temperature causes atomic diffusion and allows for the redistribution of alloying elements, including manganese, within the weldment. Therefore, manganese segregation was reduced after heat treatment because this diffusion process helps to homogenize the microstructure and promote the formation of a more uniform distribution of manganese and other elements. 

Figure 3 illustrates the relationship between Mn content and the calculated SFE of high-Mn steel at different temperatures. Based on a previous research paper [19], Figure 3a specifically illustrates the results of quantifying the Mn content before and after being heat-treated at 1000 °C. In samples as-welded with a high Mn content, changes in the Mn concentration were observed between the interdendritic and dendritic regions due to Mn depletion and segregation resulting from the SAW welding process. Since the solubility of Mn in the liquid phase exceeds that in the solid phase, the interface of solid and liquid rejects Mn into the liquid [23,24,25]. Consequently, the Mn content in the interdendritic region is higher than that in the dendritic region. The Mn content in dendritic and interdendritic areas was evaluated by conducting EDS analysis and calculating the average of five measurements for each region. Before heat treatment (As-welded), the Mn content in the dendritic area was 21.462 wt.% and the Mn content in the interdendritic area was 29.264 wt.%. As a result, there was a difference of 7.802 wt.% due to Mn-segregation. After heat treatment (1000 °C), the Mn content in the dendritic area rose to 23.302 wt.%, while the Mn content in the interdendritic region dropped to 25.978 wt.%. After heat treatment, there was an increase in the Mn of the dendritic area and a decrease in the interdendritic area, implying that heat treatment led to the diffusion of Mn in the weldment.

Figure 3b shows the SFE of welded high-Mn steel in the dendritic and interdendritic regions as a function of temperature, and it was calculated by Curtze’s model [4]. Generally, the SFE is influenced by factors such as chemical composition and temperature [3,4,5,33]. In this study, the SFE was calculated based on the results of the EDS analysis (Figure 3a) and displayed as a function of temperature. The as-welded specimen showed an SFE of approximately 25 mJ/m^2^ for the dendrite area and 35 mJ/m^2^ for the interdendritic area at room temperature. The reason is that the interdendritic region had a higher amount of Mn, which helps to maintain the austenite phase, resulting in a higher SFE in that area compared to the dendritic region [21,34]. After heat treatment, the SFE decreased to around 27 mJ/m^2^ for the dendrite region and 31 mJ/m^2^ for the interdendritic region, indicating lower values compared to the as-welded specimen. When SFE is between 20 mJ/m^2^ and 40 mJ/m^2^, it exhibits a twinning mechanism. Thefore, both specimens demonstrated deformation twinning as the prevailing deformation mechanism at room temperature. In addition, the SFE decreased with decreasing temperature in the as-welded and W1000 specimens. As a result, the SFE of the dendrite and interdendritic regions of the welded specimen decreased to about 15 mJ/m^2^ and 28 mJ/m^2^ when the temperature reached cryogenic temperature (−160 °C). In particular, the dendritic region belongs to the region where martensite transformation is possible. Similarly, the SFE of the dendrite and interdendritic regions of the W1000 specimen was approximately 19 mJ/m^2^ and 23 mJ/m^2^, respectively, which was lower than the SFE of the as-welded specimen. Some studies have shown that martensite transformation as well as twinning mechanisms may occur instead of plastic deformation when the SFE is below 20 mJ/m^2^ [3,4]. Therefore, it can be expected that martensite may be formed when deformed at cryogenic temperatures. The SFE has a significant impact on understanding deformation behavior since it is influenced by chemical composition and temperature.

### 3.2. The Mechanical Characteristics of Welded High-Mn Steels

Figure 4a presents the engineering stress–strain behavior for the as-welded specimen and the W1000 specimen at room and cryogenic temperatures. The as-welded specimen had a yield strength (YS) of 384 MPa, an ultimate tensile strength (UTS) of 656 MPa, and an elongation of 72% at room temperature. As the temperature decreased to cryogenic temperatures, the YS and UTS increased to 510 MPa and 935 MPa, respectively, and the elongation decreased to 43%. This phenomenon can be attributed to the development and proliferation of twins within the high-Manganese steel, which serve as obstacles to the movement of dislocations, thereby leading to enhanced strength. During tensile loading at cryogenic temperatures, high-Mn steel experiences significant deformation, leading to the formation of twins. These deformation twins impede the movement of dislocations, enhancing the material’s resistance to deformation and promoting strain hardening. The twin boundaries serve as a strong barrier to the motion of dislocation, impeding the propagation of dislocations and enhancing the material’s strength. Moreover, based on the previous SFE analysis (Figure 3b), the as-welded specimen belongs to the martensite transformation region with an SFE below 20 mJ/m^2^ at cryogenic temperatures. Consequently, the creation of deformation twins and martensite during deformation at extremely low temperatures resulted in an increase in the UTS from 656 MPa to 935 MPa. Conversely, the reduction in elongation from 72% to 43% in the as-welded specimens can be attributed to the development of martensite. 

In the case of heat treatment at 1000 °C, the W1000 specimen exhibited improved UTS and elongation compared with the as-welded specimen at room temperature. The YS and UTS were 457 MPa and 687 MPa, respectively, with an elongation of 91%. However, in the case of the W1000 specimen, when the temperature was lowered from room temperature to cryogenic, the YS decreased to 443 MPa, the UTS decreased to 561 MPa, and the elongation was reduced to 11.8%. In contrast to the as-welded specimens, the YS of the heat-treated samples increased from 405 MPa to 443 MPa. However, both the tensile strength and elongation decreased from 687 MPa to 561 MPa and from 91% to 11.8%, respectively. According to the result of the calculated SFE (Figure 3), the SFE of W1000 was less than 20 mJ/m^2^ at cryogenic temperature, which corresponds to the martensite transformation region. Therefore, martensite may be formed in the W1000 specimen during deformation at cryogenic temperatures [3,4]. As a result, the YS increases and the elongation decreases due to the transformation to martensite when deformed at cryogenic temperature, but the decrease in UTS compared to room temperature is considered to be due to other factors.

Figure 4b illustrates the strain hardening rate of the as-welded and W1000 specimens at both room temperature and cryogenic temperature. During the initial deformation stage, as the strain increases, the strain hardening rate decreases due to dynamic recovery through the annihilation of dislocation. The heat-treated specimen (W1000_RT) had a higher strain and strain hardening rate compared to the as-welded specimen (As-welded_RT), which means it underwent more deformation. In the case of as-welded_RT specimens, non-homogeneous microstructures with a relatively high percentage of Mn-segregation can act as stress concentrators, leading to premature local failure and reduced strain hardening. In contrast, a heat-treated specimen (W1000_RT) with a homogeneous microstructure lacks such stress concentrators, allowing for a more uniform distribution of stress and strain and thus a higher strain hardening rate. For the cryogenic temperature, the as-welded specimen showed the highest strain hardening rate owing to martensite transformation; however, the W1000 specimen had the lowest rate and strain due to poor strain hardening capacity [20]. Necking rarely occurs due to embrittlement and stress concentration at martensite intersections, leading to immediate fracture. This suggests that the deformation behavior changed from the twinning mechanism to martensite formation (γ → ε or γ → ε → α′-martensite) during plastic deformation at cryogenic temperature, related to the TRIP deformation mechanism [32,35,36]. The ε or α′-martensite typically exhibits a higher hardness and strength compared to the austenite phase. This enhanced strength can promote work hardening, which occurs through dislocation multiplication, interaction, and entanglement. The existence of ε or α′-martensite can facilitate more effective work hardening, resulting in a higher strain hardening rate.

The XRD patterns before and after tensile deformation at room temperature and cryogenic temperature are presented in Figure 5. Before tensile deformation, the XRD patterns showed only austenitic peaks, specifically (111), (200), (220), (311), and (222), in both the as-welded and W1000 specimens, as shown in Figure 5a. Welded high-Mn steels with 24 wt.% Mn or higher exhibit a completely austenitic structure at room temperature. As evident from Figure 5b, only austenite peaks persisted in the as-welded and W1000 specimens following tensile deformation at room temperature, confirming the activation of the TWIP mechanism. This suggests that twinning occurs instead of the formation of martensite when deformed at room temperature. Figure 5c exhibited the XRD patterns of the as-welded and W1000 specimens in tensile fracture at cryogenic temperature. In contrast to the observations at room temperature, the XRD patterns of the as-welded specimen deformed at cryogenic temperature reveal the existence of ε and α′-martensite. It could be confirmed that (111), (200), (220), (311) and (222) of austenite peaks, and (101) of ε-martensite and (211) of α′-martensite were obtained in an as-welded specimen. The calculated SFE results (as seen in Figure 3) explain that the SFE of the as-welded specimen is within the range of 15–28 mJ/m^2^, which belongs to the TRIP mechanism. Therefore, when the as-welded specimen is deformed at cryogenic temperatures, it can be observed that not only the twinning mechanism but also the TRIP mechanism occur. In the case of the W1000 specimen, the austenite peaks of (111), (200), (220), (311), (222), and the ε-martensite of (101) were obtained. The calculated SFE for the W1000 specimen is within the range of 19–23 mJ/m^2^, which implies that it belongs to both the twinning and TRIP mechanisms. Studies have shown that α′-martensite is formed at a calculated SFE of 18 mJ/m^2^ or less, thus it was confirmed to be present in the as-welded specimen [4].

EBSD observations such as inverse pole figure (IPF) maps, phase maps, and grain boundary (GB) maps were performed on tensile-fractured samples to investigate the deformation mechanisms of as-welded and W1000 specimens after tensile deformation at cryogenic temperatures, as shown in Figure 6. The deformed microstructure of the as-welded specimen was found to consist of a dendrite structure, with deformation twins forming within the grains (as seen in Figure 6a). It is evident that both deformation twins and ε as well as α′-martensite (indicated by yellow and red, respectively) formed within the grains after the tensile deformation at cryogenic temperature, as shown in Figure 6b. Generally, the transformation from γ to ε occurs through the formation of alternate layers of stacking faults (SFs) in the (111) planes of austenite [37]. The formed ε-martensite grows until it reaches the adjacent grain boundary. Especially, α′-martensite can be formed directly from austenite or at the intersections of ε-martensite [11]. In Figure 6c, the low angle grain boundaries (LAGBs) (green lines) can be seen to be distributed inside the austenitic grains or along the grain boundary, but not within the martensite. Additionally, it is confirmed that deformation twins (red lines) form within the grains during plastic deformation. The tensile test results presented in Figure 4 suggest that the increase in YS and UTS and the decrease in elongation in the as-welded specimen can be attributed to the creation of deformation twins and martensite (ε and α′) during deformation at cryogenic temperatures.

Figure 6d to f presents the IPF maps of the fractured tensile specimen that was heat-treated at 1000 °C. After tensile testing at cryogenic temperatures, deformation twins and martensite were observed to have formed within the grains, as shown in Figure 6d. In contrast to the result of the as-welded specimen, it was confirmed that only ε-martensite (indicated by yellow) generated within the grains, and α′-martensite was not observed (Figure 6e). Additionally, it can be observed that ε-martensite grows in the opposite direction from the grain boundary. In Figure 6f, deformation twins and LAGBs are formed along grain boundaries, but much less than in the as-welded specimen. This can be explained in relation to the very low elongation at cryogenic temperatures. Deformation twins are generally considered to be an important cause of material deformation, which contributes to the improvement of strength and elongation of materials due to dynamic grain refinement. However, LAGBs or dislocations can decrease due to the rearrangement of atoms within the grains after heat treatment, and materials with low LAGBs or dislocations are difficult to deform and have increased brittleness, which can lead to easy failure of the material without deformation. Therefore, as can be seen in Figure 6f, the movement of dislocations in the material decreases at cryogenic temperatures, and the absence of sufficient LAGBs and deformation twins contributes to the decrease in strength and elongation [38,39]. The EBSD analysis findings align well with the results obtained from SFE and XRD analysis conducted following tensile testing at cryogenic temperatures, providing evidence for the validity of the observations.

Figure 7 presents the bright field (BF) images and corresponding selected area diffraction patterns (SADPs) of the as-welded and W1000 specimens after tensile deformation at cryogenic temperature. Figure 7a displayed a BF image of the tensile fractured as-welded specimen with a complex phase with austenite and martensite due to severe deformation. The corresponding SADP in Figure 7b shows the existence of austenite, twin, ε, and α′-martensite in the regions indicated by the magenta dash line, green dash line, yellow dash line, and red dash line, respectively. This indicates that deformation twin and martensite (ε and α′) are formed when deformed at cryogenic temperatures. In addition, it was confirmed that α′-martensite has a Kurdjumov–Sachs (K–S) orientation relationship ({111}_γ_ || {110}_α_, <110>_γ_ || <111>_α_) with an austenite matrix [40,41] and ε-martensite displays the Shoji–Nishiyama (S–N) orientation relationship ({111}_γ_ || {0001}_ε_, <10 $\bar{1}$>_γ_ || <11 $\bar{2}$0>_ε_) with the austenite matrix [42,43]. According to some research [10,11], ε-martensite nucleates at the intersection of deformation twins, whereas α′-martensite forms either at the intersection of deformation twins or within existing ε-martensite plates. So, it was confirmed that ε and α′-martensite transformations as well as deformation twins are formed when SFE decreases below 20 mJ/m^2^. In accordance with the previous results (Figure 2 and Figure 3), the existence of Mn segregation during the solidification of the weldment was attributed to the solubility difference between the solid and liquid phases. The difference in Mn contents causes local SFE deviations, resulting in different deformation mechanisms depending on the dendrite and interdendritic regions. The research findings indicate that at cryogenic temperatures, deformation results in the creation of deformation twins and ε and α′-martensite. It was particularly evident in welded samples that had significant variations in Mn content. The reduction in SFE observed in the dendrite region, which had a lower concentration of manganese, contributed to the ε and α′-martensite formations.

Figure 7c displays a BF image of the W1000 specimen with two different deformation twin systems. The corresponding SADP in Figure 7d shows the austenite matrix and deformation twin. In the case of W1000 specimens, α′-martensite was not observed after cryogenic deformation, whereas only deformation twins were presented. However, it is evident that ε-martensite definitely existed in the W1000 specimen through EBSD analysis. The EDS results (Figure 3) indicated that heat treatment led to a reduction in Mn-segregation between the dendrite and interdendritic regions in the W1000 specimens. In the dendrite region, the SFE increased to 19–23 mJ/m^2^ as Mn content rose due to diffusion. Therefore, the heat-treated welded specimen (W1000) has a low probability of the occurrence of α′-martensite as a deformation mechanism at cryogenic temperatures. Through TEM and EBSD analysis, it was confirmed that α′-martensite did not occur, and deformation twins and ε-martensite were formed. As a result, it can be confirmed that the XRD results after the cryogenic tensile deformation agree well with the EBSD and TEM analysis results. 

Figure 8 shows the SEM fractography of the as-welded specimen and the heat-treated specimen (W1000) tensile tested at cryogenic temperature, and the fracture surfaces in both the as-welded and W1000 specimens exhibited dimple fracture, which is a feature of ductile fracture mode. In the as-welded specimen, it typically shows ductile fracture with fine dimples, which is commonly observed in the weld zone, as shown in Figure 8a. Also, the size and depth of the fine dimples in the as-welded specimen were smaller than those in the W1000 specimen (Figure 8b). When the as-welded specimen deformed at cryogenic temperature, not only deformation twins but also martensite formed (Figure 5a–c). It is especially considered that the development of fine dimples is associated with the activation of deformation twins [17]. However, the W1000 specimen exhibited a mixture of fine dimples and quasi-brittle fracture, and more quasi-cleavage fracture was observed, as shown in Figure 8b. This is related to the formation of deformation twins and martensite (Figure 6d–f), but fewer deformation twins were partially formed within the grains, unlike the as-welded specimen. Therefore, it is indicated that the reduction in strength and elongation in the W1000 specimen is attributed to the martensite transformation during tensile deformation at cryogenic temperature.

Figure 9a presents the Charpy impact absorbed energy for the as-welded specimen and the W1000 specimen at room and cryogenic temperatures. The Charpy impact energy at room temperature of the welded specimen was 143 J, and that of the W1000 specimen was 139 J. The as-welded specimens had slightly higher Charpy impact absorbed energy than the W1000 specimens, but there was no significant difference. At cryogenic temperature (−160 °C), the impact energies for the as-welded and W1000 specimens were about 123 J and 112 J, respectively. Therefore, the impact energies of the as-welded and W1000 specimens were lower at cryogenic temperatures compared to those at room temperature. At cryogenic temperatures, the mobility of twin boundaries and the slip system decreases, leading to reduced energy absorption capability. The reduction in impact energy is mainly due to the restricted slip system movement and the inability of twins to effectively accommodate deformation [44]. Consequently, at cryogenic temperatures, the impact energy decreases due to the increased difficulty in activating twinning and slip systems in the materials. 

Figure 9b,c exhibited the load–displacement curves for as-welded and W1000 specimens acquired through instrumented Charpy impact testing at both room and cryogenic temperatures. In this study, the crack initiation energy (E_i_) and the crack propagation energy (E_p_) were determined by calculating the area under the load–displacement curve, with the maximum load (P_max_) serving as the reference point. The summarized results related to the instrumented data are presented in Table 1. The Pmax of as-welded and W1000 specimens at room temperature was approximately 9.4 kN and 10.1 kN, respectively, and that of the as-welded and W1000 specimens at cryogenic temperatures was approximately 13.2 kN and 13 kN, respectively (Table 1). Therefore, the Pmax value was greater at cryogenic temperatures compared to room temperature. At cryogenic temperatures, materials, including high-Mn steel, generally exhibit increased stiffness. The lower temperatures reduce the thermal activation of dislocations, making it more difficult for plastic deformation to occur. This reduced ductility causes the material to experience less deformation and higher stresses during impact, leading to a higher P_max_. The increased material stiffness results in higher energy being required to initiate a crack, leading to higher Ei. In the case of Ep, high-Mn steel at cryogenic temperatures tends to increase brittleness. The reduced mobility of dislocations limits the material’s ability to undergo plastic deformation, and failure occurs primarily through a brittle mechanism. As a result, when a crack is initiated in a material, it propagates more readily through the brittle structure, resulting in a lower crack propagation energy (E_p_) than at room temperature.

To understand the phase transformations occurring during the Charpy impact test at both room and cryogenic temperatures, XRD analysis was carried out on the deformed Charpy V-notched specimens, as depicted in Figure 10. The XRD patterns for the as-welded and W1000 specimens were exhibited after impact testing at room temperature, as shown in Figure 10a. After impact deformation from dynamic loading, the as-welded and W1000 specimens exhibit only austenite peaks related to the (111), (200), (220), (311), and (222) planes. The absence of martensite transformation during dynamic loading at room temperature indicates the ability of welded high-Mn steels to maintain their austenitic structure, a characteristic attributed to the TWIP mechanism. 

Figure 10b presents the XRD patterns for the as-welded and W1000 specimens after Charpy impact testing at cryogenic temperatures. The results were consistent with those obtained at room temperature, as the deformed welded high-Mn steels exhibited completely austenite peaks. The calculated SFE for the as-welded specimen at cryogenic temperature is approximately 15 mJ/m^2^, while for the W1000 specimen, it is approximately 19 mJ/m^2^. These values indicated that a phase transformation to martensite could be feasible, but it did not take place. Despite this, the tensile test results mentioned above showed that both as-welded specimens and W1000 specimens underwent martensitic transformation during tensile loading (static deformation) at cryogenic temperatures. Intriguingly, the deformation behavior of welded high-Mn steels is influenced by both temperature and loading rate. A detailed analysis of deformation mechanisms observed in tensile and Charpy impact tests carried out at various temperatures and loading rates is presented in the subsequent section.

Figure 11a,b show the EBSD analysis of the as-welded and W1000 specimens after impact fracture at room temperature, respectively. The cross-section of the fractured, as-welded specimen, as shown in Figure 11a, exhibits deformation twins and a high proportion of LAGBs distributed within the grains, but martensite was not observed. In contrast to the as-welded specimen, the fractured W1000 specimen had a low proportion of LAGBs in its cross-section, with the absence of martensite and deformation twins, as depicted in Figure 11b. Subjected to dynamic loading conditions at room temperature, rapid deformation does not lead to the martensite transformation in the welded high-Mn steel, which can be attributed to its TWIP behavior, maintaining a fully austenitic phase. In addition, as mentioned in the XRD analysis result (Figure 10b), it can be observed that the result of XRD analysis after the Charpy impact deformation for both as-welded and W1000 specimens exhibited only austenite peaks at room temperature, which was consistent with the EBSD findings. Figure 11c,d shows the EBSD of the as-welded and W1000 specimens after Charpy impact fracture at cryogenic temperature, which reveals the deformation behavior observed around the cross-sectional area adjacent to the crack path. The crack propagation region of the as-welded specimen (Figure 11c) exhibits a distribution of LAGBs along grain boundaries and deformation twins within grains. The W1000 specimen exhibits a low proportion of LAGBs and a high density of deformation twins within the grains, as shown in Figure 11d. Additionally, at cryogenic temperatures, martensite transformation did not occur after dynamic loading. Especially, the calculated SFE values of the as-welded and W1000 specimens were about 15 mJ/m^2^ and 19 mJ/m^2^ at cryogenic temperature, respectively. It is evident that the formation of martensite could have been possible, but it did not occur. 

The instrumented data can be utilized to estimate the dynamic yield strength through a well-established relationship, according to some studies. While the loading modes in the tensile and Charpy impact tests are different, it is still valuable to compare the results of the two tests due to the significant difference in loading rate. A comparison of the tensile and Charpy impact test findings showcased the existence of martensite in the fractured tensile specimen, in contrast to its absence in the fractured Charpy impact specimen. The strain rate of the tensile testing was performed at 10^−3^/s, while impact testing is performed with a loading rate of 5.424 m/s. Consequently, the pendulum in the Charpy impact test imparts a significantly faster loading force on the specimen compared to the crosshead in the tensile test. In accordance with some research [45,46], self-heating effects occur during plastic deformation at high loading speeds owing to the adiabatic effect, leading to heat generation. Shifting from quasi-static loading to dynamic loading typically causes a substantial temperature rise in a specimen that fractures during deformation [32,47]. When subjected to rapid dynamic loading at 10/s and 100/s strain rates, the temperature of fractured TWIP steel specimens exceeds 120 °C, prompting a change in the material’s deformation mechanisms [48]. During the Charpy impact test, the material undergoes rapid deformation and experiences high strain rates, leading to significant adiabatic heating. The impact test pendulum’s velocity was approximately 5000 mm/s, which corresponds to strain rates of 625/s. This strain rate is much higher compared to the strain rate used in a tensile test (10^−3^/s). When the steel is subjected to deformation at a strain rate of 600/s, the adiabatic effect causes the specimen’s temperature to increase by approximately 160–230 °C in the deformation region [49]. Consequently, during the Charpy impact test conducted at cryogenic temperatures, the temperature in the deformation region may have risen to approximately 0–70 °C. By referring to our calculated SFE (Figure 3), the SFE can increase from 15 mJ/m^2^ (As-welded) and 19 mJ/m^2^ (W1000) to 39 mJ/m^2^ and 36 mJ/m^2^, respectively. As a result, when deformed at cryogenic temperatures and high strain rates, the primary deformation mechanism shifts from martensitic transformation to twinning. An increase in temperature caused by the adiabatic heating effect leads to an increase in the SFE. Consequently, this shift in SFE under dynamic loading influences the deformation behavior, whether it involves twinning or dislocation glide [47]. Especially at cryogenic temperatures, it is evident that an increase in the loading rate triggers a shift from TRIP to TWIP deformation behavior.

## 4. Conclusions

This study discusses the influence of heat treatment and deformation temperatures on the mechanical properties of the weldment of high-Mn steel before and after PWHT. It was found that the difference in Mn content between the dendritic and interdendritic regions decreased after heat treatment and that the SFE was calculated based on EDS analysis. At room temperature, the SFE was over 20 mJ/m^2^, corresponding to the TWIP mechanism, but at cryogenic temperatures, it decreased below 20 mJ/m^2^, indicating the martensitic transformation region. The tensile test at cryogenic temperatures showed that the YS and UTS of the as-welded specimen increased but the elongation decreased, while both strength and elongation of the W1000 decreased. The instrumented Charpy impact test revealed that the as-welded specimen exhibited marginally higher absorbed energy than the W1000 specimen at cryogenic temperatures, while demonstrating superior resistance to crack initiation. It was found that the tensile deformation mode at cryogenic temperature exhibited the TRIP; the deformation mechanism is influenced by the SFE, but the impact deformation mode still maintains the TWIP mechanism regardless of the SFE due to the rapidly increasing temperature after impact loading. This study analyzes the deformation behavior of welded high-Mn steel in cryogenic environments, providing insights into deformation mechanisms at cryogenic temperatures. It also clarifies the effects of PWHT and offers valuable guidance for the design and fabrication of high-Mn steel for cryogenic industrial applications. Therefore, this research makes a significant scientific contribution to supporting practical applications in this field.

## Figures and Tables

**Figure 1 materials-17-04159-f001:**
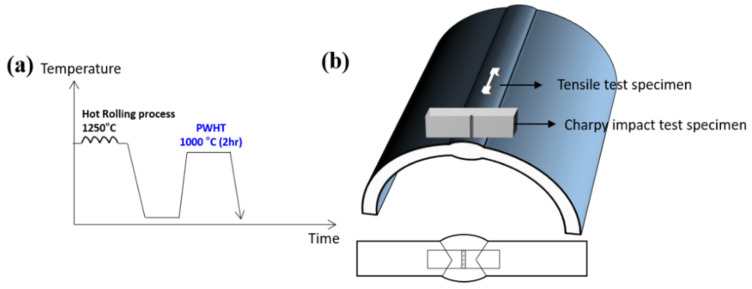
(**a**) A schematic diagram of the PWHT process and (**b**) location of Charpy impact test specimen and tensile test specimen in welded high-Mn steel.

**Figure 2 materials-17-04159-f002:**
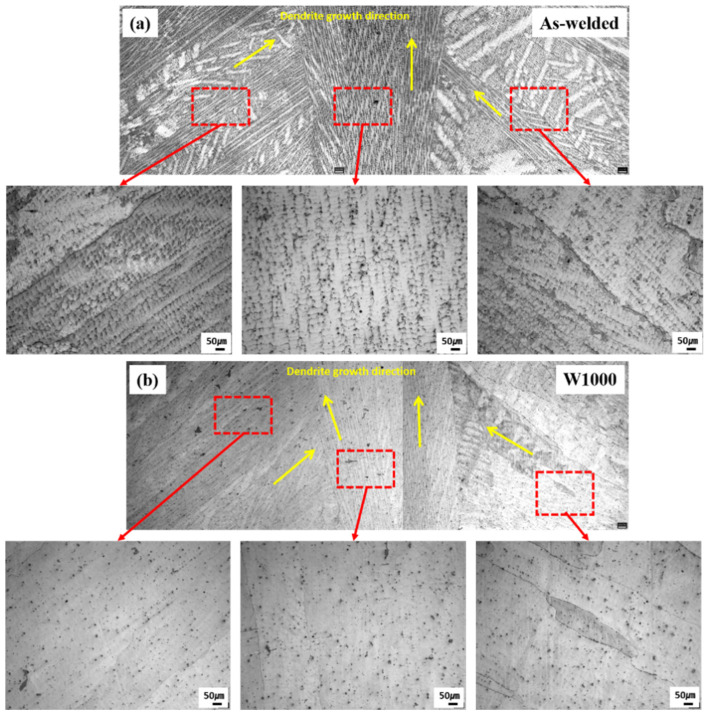
The microstructure of welded high-Mn steel before and after heat treatment: (**a**) before heat treatment (as-welded) and (**b**) heat-treated at 1000 °C (W1000); Red box indicates the enlarged area of the weldment. Yellow arrows indicate dendrite growth direction.

**Figure 3 materials-17-04159-f003:**
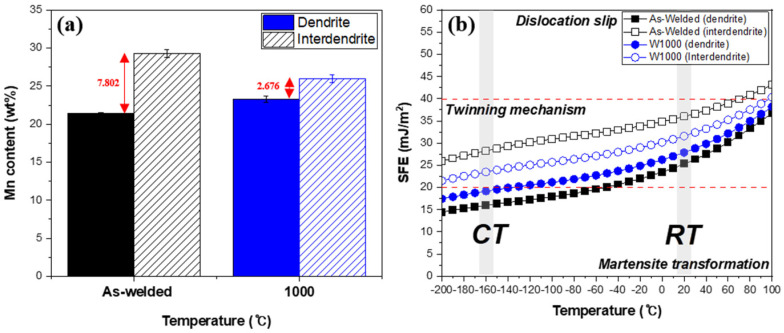
(**a**) EDS results for Mn content of welded high-Mn steel in dendrite and interdendritic areas before and after heat-treatment at 1000 °C. Each red arrow represents the difference in Mn content in the dendrite and interdendrite regions before and after heat treatment. (**b**) SFE of welded high-Mn steel in dendrite and interdendritic regions as function of temperature. The SFE was calculated by Curtze’s model.

**Figure 4 materials-17-04159-f004:**
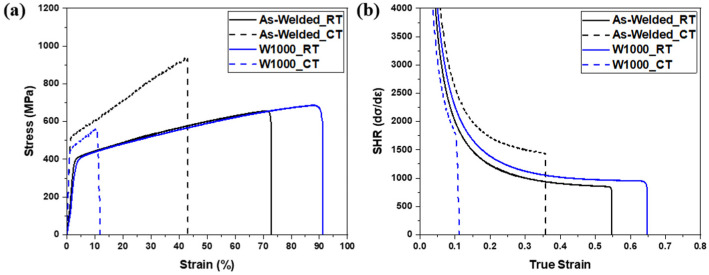
(**a**) Engineering stress–strain curves and (**b**) strain hardening rate of as-welded and W1000 specimens at room temperature and cryogenic temperature.

**Figure 5 materials-17-04159-f005:**
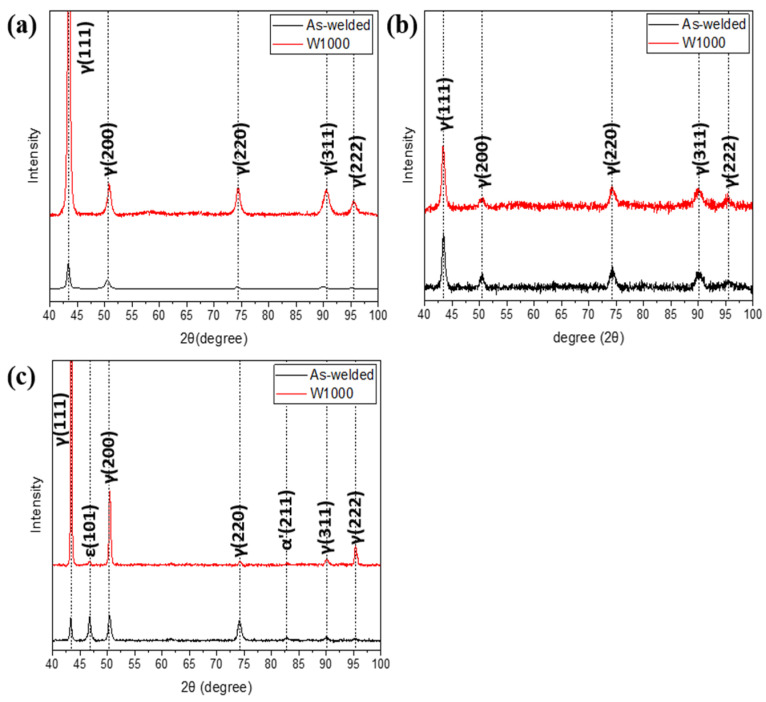
X-ray diffraction (XRD) patterns for the as-welded and W1000 specimens before and after tensile deformation at the different temperatures: (**a**) Before deformation; (**b**) Tensile specimen fractured at room temperature (RT); and (**c**) Tensile specimen fractured at cryogenic temperature.

**Figure 6 materials-17-04159-f006:**
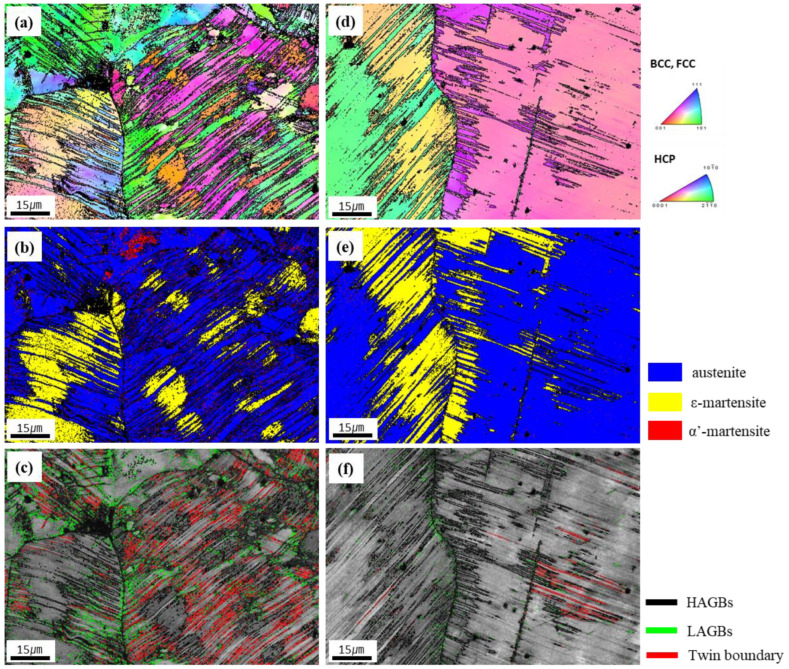
EBSD inverse pole figure (IPF) map, phase map, and grain boundary (GB) map of the tensile fractured specimens at cryogenic temperature for the as-welded and W1000 specimens: (**a**) IPF map of as-welded, (**b**) phase map of as-welded, (**c**) GB map of as-welded, (**d**) IPF map of W1000, (**e**) phase map of W1000, and (**f**) GB map of W1000.

**Figure 7 materials-17-04159-f007:**
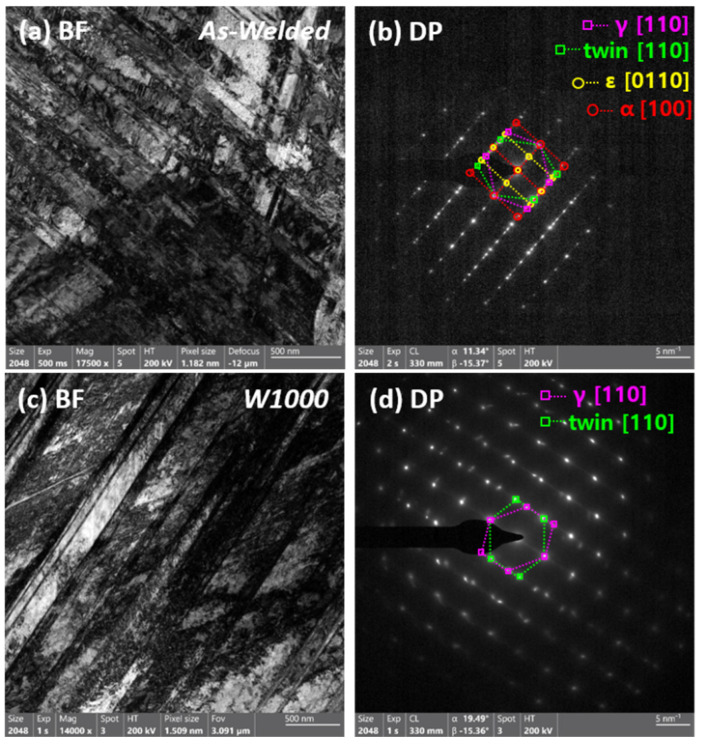
Transmission electron micrograph of the welded high-Mn steel deformed at cryogenic temperature (−160 °C): (**a**) bright field image of as-welded specimen; (**b**) diffraction patterns of austenite (magenta line), twin (green line), ε and α′-martensite (yellow line and red line, respectively); (**c**) BF image of heat-treated specimen (W1000); and (**d**) diffraction patterns of austenite and twin.

**Figure 8 materials-17-04159-f008:**
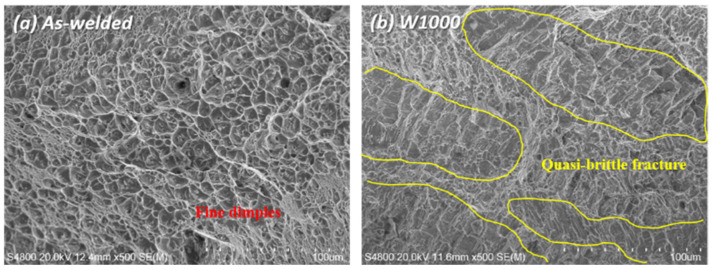
SEM fractography of as-welded specimen and W1000 specimen tensile tested at cryogenic temperature: (**a**) As-welded specimen and (**b**) W1000.

**Figure 9 materials-17-04159-f009:**
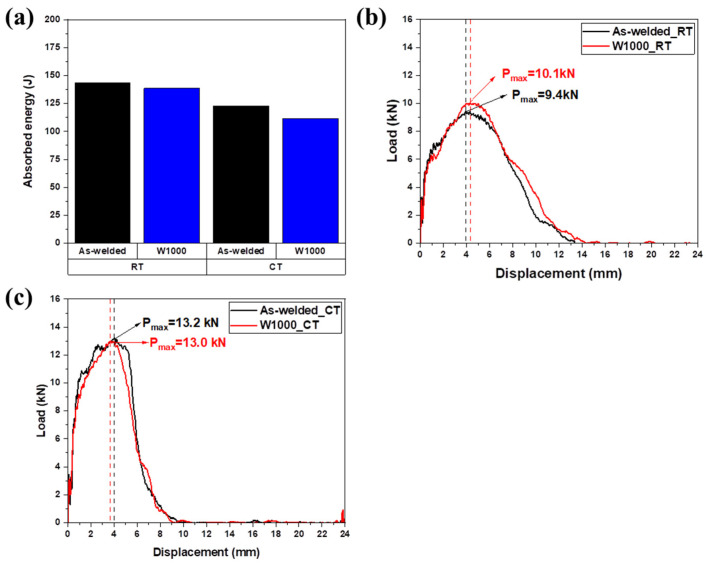
Charpy impact absorbed energy and the load–displacement curves for as-welded and W1000 specimens acquired through instrumented Charpy impact test at room temperature (20 °C) and at cryogenic temperature (−160 °C): (**a**) Charpy impact absorbed energy, load–displacement curves (**b**) at room temperature (RT), and (**c**) at cryogenic temperature (CT).

**Figure 10 materials-17-04159-f010:**
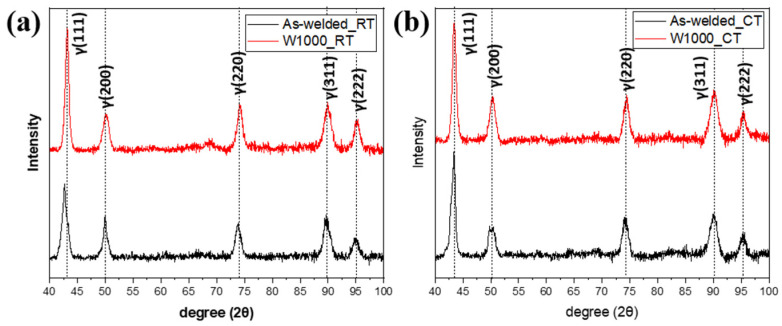
X-ray diffraction (XRD) patterns for the as-welded and W1000 specimens before and after impact deformation at the different temperatures: (**a**) Charpy impact specimen fractured at room temperature (RT) and (**b**) Charpy impact specimen fractured at cryogenic temperature (CT).

**Figure 11 materials-17-04159-f011:**
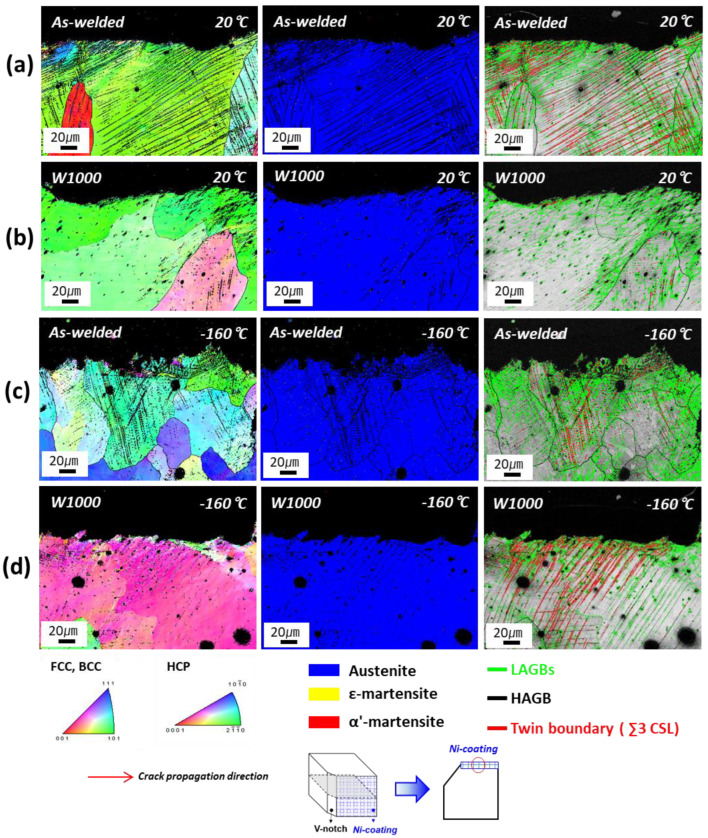
EBSD inverse pole figure (IPF) map, phase map, and grain boundary (GB) map of welded high-Mn after Charpy impact fractured at room temperature (20 °C) and cryogenic temperature (−160 °C): (**a**) As-welded_RT, (**b**) W1000_RT, (**c**) As-welded_CT, and (**d**) W1000_CT.

**Table 1 materials-17-04159-t001:** Charpy impact absorbed energy and instrumented data at room temperature (20 °C) and cryogenic temperature (−160 °C) of welded high-Mn steels.

Samples	Temperature (°C)	P_max_(kN)	Total Absorbed Energy (J)	Crack Initiation Energy (E_i_, J)	Crack Propagation Energy (E_p_, J)
As-weld_RT	20	9.4	143	57	86
W1000_RT		10.1	139	57	82
As-weld_CT	−160	13.2	123	74	49
W1000_CT		13	112	61	51

## Data Availability

Data are contained within the article.

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
