# Peer review of "Temperature Effect on Deformation Mechanisms and Mechanical Properties of Welded High-Mn Steels for Cryogenic Applications"

_materials, 2024, doi:10.3390/ma17164159_

Round 1

Reviewer 1 Report

Comments and Suggestions for Authors

The submitted manuscript discusses the temperature effect on deformation mechanisms and mechanical properties of welded high-Mn steels for Cryogenic Applications. The manuscript topic highlights an importanr aspect related to steel alloys for crygenic applications. The manuscript falls well within the journal theme. The writing language is good and very minor English editing is required. The overall structure of the research is perfect. The methods are well chosed and described adequately. The results are fairly discussed and supported with experimental evidences. The conclusion is very related to the entire manuscript contents. However, very minor corrections are required:

1- The initials of "Crygenic Applications" in the title should not be capitalized. 

2- The temperature unit in Line 150 must be correctly typed by adding the degree symbol.

3- The units after numbers are inconsistent. Sometimes authors do not leave a space between them and other times they do. All numbers and units style must be consistent in the entire manuscript.

Author Response

Reviewer #1 : 

We are most grateful for the reviewers' helpful and detailed comments. We agree that the original manuscript should be revised to address the justified comments given by the reviewers. Accordingly, the revised manuscript has been systematically improved with new information and additional interpretations. We enclose the correspondingly revised version of the paper. Below, we describe how the paper was modified to address and comply with the reviewers’ comments. We sincerely hope that these changes and additional explanations render the article acceptable for publication in Materials. All the amendments are outlined below in more details.

  1. The initials of "Crygenic Applications" in the title should not be capitalized.

Thank you for your valuable comment. We have corrected the capitalization of the title from 'Cryogenic Application' to 'cryogenic application'."

  1. The temperature unit in Line 150 must be correctly typed by adding the degree symbol.

Thank you for pointing this out. We have revised ‘85oC’ and ‘-163 oC’ to ’85 °C’ and -163°C’.

  1. The units after numbers are inconsistent. Sometimes authors do not leave a space between them and other times they do. All numbers and units style must be consistent in the entire manuscript.

 We thank for this suggestion. We modified the style to include spaces between all numbers and units and made corrections throughout the manuscript.

Reviewer 2 Report

Comments and Suggestions for Authors

In order to examine the effects of post-weld heat treatment on the mechanical properties of welded high-Mn steels at room and cryogenic temperatures, a suite of analytical techniques, including XRD, EBSD and TEM were employed to analyze the microstructure of welded high-Mn steel before and after heat treatment. The results obtained have high engineering value. The manuscript is considered acceptable in its present state, however, the author needs to clarify the following points in the manuscript:

(1)  What type of flux should be used for welding?

(2)  Why choose to heat to 1000℃ during post-welding heat treatment?

(3)  The abstract needs to be further simplified.

Author Response

Reviewr #2 :

We are most grateful for the reviewers' helpful and detailed comments. We agree that the original manuscript should be revised to address the justified comments given by the reviewers. Accordingly, the revised manuscript has been systematically improved with new information and additional interpretations. We enclose the correspondingly revised version of the paper. Below, we describe how the paper was modified to address and comply with the reviewers’ comments. We sincerely hope that these changes and additional explanations render the article acceptable for publication in Materials. All the amendments are outlined below in more details.

  1. What type of flux should be used for welding?

    A SAW flux with a submerged wire (POS-CF1) and the baked type flux for cryogenic high-Mn steel manufactured by Poswelding (Republic of Korea) was used in this study. We have added this information to the manuscript in materials and method section.

    “A submerged wire (POS-CF1) with the baked type flux for cryogenic high-Mn steel manufactured by Poswelding (Republic of Korea) was used in this study.”

  1. Why choose to heat to 1000℃ during post-welding heat treatment?

     We are much grateful for these comments. According to a previous study, we conducted an analysis of the microstructure and mechanical properties after post-weld heat treatment at 800, 900, and 1000 °C [19]. We observed that Mn segregation in the dendritic and interdendritic regions decreased after heat treatment, with the smallest difference occurring at 1000°C. Therefore, since the stacking fault energy (SFE) varies with Mn content and temperature, this study aimed to investigate the deformation mechanism of the weldment at cryogenic temperatures, where high-manganese steel is primarily used, focusing on the post-weld heat treatment temperature of 1000°C

    1. Park, M.; Kang, M.; Park, G.-W.; Jang, G.; Kim, B.; Kim, H.C.; Jeon, J.B.; Kim. H.; Kwon, S.-H.; Kim, B.J. The effects of post weld heat treatment for welded high-Mn austenitic steels using the submerged arc welding. J. Mater. Res. Techonol. 2022, 18, 4497-4512.

  1. The abstract needs to be further simplified.

Thank you for your valuable comments and we agree with your opinion. The number of words in the abstract was further simplified and revised from 255 words to 217 words.

Reviewer 3 Report

Comments and Suggestions for Authors

Review

The paper under review describes a study that deals with the effect of heat treatment and deformation temperatures on the mechanical properties of denerite and the interdendritic region. It was found that the difference in Mn content between the dendritic and interdendritic regions decreased after heat treatment and that the SFE was calculated based on EDS analysis.

In this study, submerged arc welding (SAW) was applied to fabricate thick pipes for cryogenic industry applications. However, SAW welding of high-Mn steels can pose challenges such as uneven distribution of manganese and a large weldment.

I basically agree with the formulated and stated conclusions in the post. In my opinion, the analysis and clarification of the presented research brings knowledge that can be generally formulated and is a good starting point for further research. The text is clear and well organized, so that the reader is very well versed in the described issue.

1. The process that is analyzed in the article and presented by the authors is described correctly and well. The authors have well incorporated and emphasized the necessary knowledge, which is an integral part of the solved issue, into the contribution. The authors provide important and necessary information for solving the problem in the text. The mentioned and presented topic is solved and analyzed by the authors using published professional sources in References, which the authors have chosen and collected for the solution. The number of articles listed in References is sufficient. As a reviewer at this point, I state that, in my opinion, the assessed contribution meets all the formal requirements for this kind of scientific articles. The text of the contribution is divided into only four chapters, which are filled with the necessary data and thus the contribution received a high informative value. The chapters guide the reader very well and describe in detail the procedures used by the authors of the paper. The division of the text chosen by the authors shows that the chapters logically and factually analyze and solve the investigated issue. The text of the submitted article is at a relatively high level from both a professional and a scientific point of view. I rate the Introduction chapter as sufficient. In this chapter, the authors analyzed the contributions of the References. This analysis is important because this chapter is the basis for all scientific contributions. Chapter 2. Materials and Methods is developed correctly and well. Chapter 3. Results is very well illustrated by the authors with high-quality images and graphs. Some images may need to be enlarged. Look further. I appreciate the detailed description of the research findings. The authors have developed this chapter very well. The stated findings are valuable contributions to this rather extensive issue. I also appreciate the precise handling of the results illustrated in Figures 2-11. This chapter is a confirmation of the authors' detailed work in their research. The assessed contribution indicates and perhaps even proves that the authors have sufficiently processed the solved issue. Conclusions – the text contains the facts found by the authors of the article. I agree with this chapter. The scientific contribution of the authors should be clearly stated. My opinion - the presented research brings new and valuable knowledge, which will become the basis for the following analysis and processing.

2. I can evaluate and summarize the chosen topic as follows: as a reviewer, I think that the given topic was chosen very appropriately and correctly. The topic is analyzed in detail and will certainly interest and stimulate the professional and scientific public. The experiments performed by the authors were performed correctly and well. The assessed contribution points out that the text of the contribution has the required expressiveness. Regarding the actual presentation of the evaluated research, which was developed by the authors, I can evaluate this fact as appropriate and correct. The contribution of the authors lies in how in detail the authors analyzed the supporting aspects of the analyzed problem. The presented research itself fits into the field of science.

3. After a detailed study, evaluation and analysis of the text of the peer-reviewed paper submitted by the authors, I became convinced that the authors are acting at a completely professional level in this important area in which they present their research together with the formulation of their findings. At this point, as a reviewer, I express the opinion that the reviewed article completely reaches the necessary scientific level. It is possible to publish it.

4. The methodology chosen by the authors for their published research is very good, and I consider and evaluate this as correct. For this kind of scientific contribution, the authors applied the correct methodology. The contribution is divided into four chapters.

5. The experimental findings published by the authors are formulated correctly, clearly and precisely. The experiments performed by the authors are presented, described and analyzed at a high scientific level. An accompanying supplement to the text of the assessed contribution is its illustration with suitable pictures, with a great explanatory value.

6. A total of 49 items are listed in the References. According to my judgment and conviction, this number of publications is sufficient for the mentioned field of research of the authors. I do not wish to add to this list.

7. I rate the images presented in the text as very appropriate and correct. There is no need to modify them, in terms of color and description. The pictures complement and help to understand the findings and the procedure for solving the problem. In their current state, they meet this requirement. See further.

8. The abstract written by the authors is good, it informs, presents and contains all the necessary data and information.

Comments, additions and questions:

1. Images 1,2,5,8 enlarge.

2. To supplement the scientific contribution of the contribution.

The post is well written and after minimal editing I recommend publishing it.

Author Response

Reviewer #3 :

We are most grateful for the reviewers' helpful and detailed comments. We agree that the original manuscript should be revised to address the justified comments given by the reviewers. Accordingly, the revised manuscript has been systematically improved with new information and additional interpretations. We enclose the correspondingly revised version of the paper. Below, we describe how the paper was modified to address and comply with the reviewers’ comments. We sincerely hope that these changes and additional explanations render the article acceptable for publication in Materials. All the amendments are outlined below in more details.

  • Images 1,2,5,8 enlarge. 

    Thank you for your comments. We agree with this comment. The sizes of Figures 1, 2, 5 and 8 have been modified to be larger.

  • To supplement the scientific contribution of the contribution.

    We are grateful for these comments. Please the manuscript has been updated with these statement.  

    “This study analyzes the deformation behavior of welded high-Mn steel in cryogenic environments, providing insights into deformation mechanisms at cryogenic temperature. It also clarifies the effects of PWHT and offers valuable guidance for the design and fabrication of high-Mn steel for cryogenic industrial applications. Therefore, this research makes a significant scientific contribution to supporting practical applications in this field.”
